# A scalable end-to-end Gaussian process adapter for irregularly sampled time series classification

**Steven Cheng-Xian Li**     **Benjamin Marlin**
College of Information and Computer Sciences
University of Massachusetts Amherst
Amherst, MA 01003
`{cxl,marlin}@cs.umass.edu`

## Abstract

We present a general framework for classification of sparse and irregularly-sampled time series. The properties of such time series can result in substantial uncertainty about the values of the underlying temporal processes, while making the data difficult to deal with using standard classification methods that assume fixed-dimensional feature spaces. To address these challenges, we propose an uncertainty-aware classification framework based on a special computational layer we refer to as the Gaussian process adapter that can connect irregularly sampled time series data to any black-box classifier learnable using gradient descent. We show how to scale up the required computations based on combining the structured kernel interpolation framework and the Lanczos approximation method, and how to discriminatively train the Gaussian process adapter in combination with a number of classifiers end-to-end using backpropagation.

## 1  Introduction

In this paper, we propose a general framework for classification of sparse and irregularly-sampled time series. An irregularly-sampled time series is a sequence of samples with irregular intervals between their observation times. These intervals can be large when the time series are also sparsely sampled. Such time series data are studied in various areas including climate science [22], ecology [4], biology [18], medicine [15] and astronomy [21]. Classification in this setting is challenging both because the data cases are not naturally defined in a fixed-dimensional feature space due to irregular sampling and variable numbers of samples, and because there can be substantial uncertainty about the underlying temporal processes due to the sparsity of observations.

Recently, Li and Marlin [13] introduced the mixture of expected Gaussian kernels (MEG) framework, an uncertainty-aware kernel for classifying sparse and irregularly sampled time series. Classification with MEG kernels is shown to outperform models that ignore uncertainty due to sparse and irregular sampling. On the other hand, various deep learning models including convolutional neural networks [12] have been successfully applied to fields such as computer vision and natural language processing, and have been shown to achieve state-of-the-art results on various tasks. Some of these models have desirable properties for time series classification, but cannot be directly applied to sparse and irregularly sampled time series.

Inspired by the MEG kernel, we propose an uncertainty-aware classification framework that enables learning black-box classification models from sparse and irregularly sampled time series data. This framework is based on the use of a computational layer that we refer to as the Gaussian process (GP) adapter. The GP adapter uses Gaussian process regression to transform the irregular time series data into a uniform representation, allowing sparse and irregularly sampled data to be fed into any black-box classifier learnable using gradient descent while preserving uncertainty. However, the

$\mathcal{O}(n^3)$ time and $\mathcal{O}(n^2)$ space of exact GP regression makes the GP adapter prohibitively expensive when scaling up to large time series.

To address this problem, we show how to speed up the key computation of sampling from a GP posterior based on combining the structured kernel interpolation (SKI) framework that was recently proposed by Wilson and Nickisch [25] with Lanczos methods for approximating matrix functions [3]. Using the proposed sampling algorithm, the GP adapter can run in linear time and space in terms of the length of the time series, and $\mathcal{O}(m \log m)$ time when $m$ inducing points are used.

We also show that GP adapter can be trained end-to-end together with the parameters of the chosen classifier by backpropagation through the iterative Lanczos method. We present results using logistic regression, fully-connected feedforward networks, convolutional neural networks and the MEG kernel. We show that end-to-end discriminative training of the GP adapter outperforms a variety of baselines in terms of classification performance, including models based only on GP mean interpolation, or with GP regression trained separately using marginal likelihood.

## 2   Gaussian processes for sparse and irregularly-sampled time series

Our focus in this paper is on time series classification in the presence of sparse and irregular sampling. In this problem, the data $\mathcal{D}$ contain $N$ independent tuples consisting of a time series $\mathcal{S}_i$ and a label $y_i$. Thus, $\mathcal{D} = \{(\mathcal{S}_1, y_1), \ldots, (\mathcal{S}_N, y_N)\}$. Each time series $\mathcal{S}_i$ is represented as a list of time points $\mathbf{t}_i = [t_{i1}, \ldots, t_{i|\mathcal{S}_i|}]^\top$, and a list of corresponding values $\mathbf{v}_i = [v_{i1}, \ldots, v_{i|\mathcal{S}_i|}]^\top$. We assume that each time series is observed over a common time interval $[0, T]$. However, different time series are not necessarily observed at the same time points (i.e. $\mathbf{t}_i \neq \mathbf{t}_j$ in general). This implies that the number of observations in different time series is not necessary the same (i.e. $|\mathcal{S}_i| \neq |\mathcal{S}_j|$ in general). Furthermore, the time intervals between observation within a single time series are not assumed to be uniform.

Learning in this setting is challenging because the data cases are not naturally defined in a fixed-dimensional feature space due to the irregular sampling. This means that commonly used classifiers that take fixed-length feature vectors as input are not applicable. In addition, there can be substantial uncertainty about the underlying temporal processes due to the sparsity of observations.

To address these challenges, we build on ideas from the MEG kernel [13] by using GP regression [17] to provide an uncertainty-aware representation of sparse and irregularly sampled time series. We fix a set of reference time points $\mathbf{x} = [x_1, \ldots, x_d]^\top$ and represent a time series $\mathcal{S} = (\mathbf{t}, \mathbf{v})$ in terms of its posterior marginal distribution at these time points. We use GP regression with a zero-mean GP prior and a covariance function $k(\cdot, \cdot)$ parameterized by kernel hyperparameters $\boldsymbol{\eta}$. Let $\sigma^2$ be the independent noise variance of the GP regression model. The GP parameters are $\boldsymbol{\theta} = (\boldsymbol{\eta}, \sigma^2)$.

Under this model, the marginal posterior GP at $\mathbf{x}$ is Gaussian distributed with the mean and covariance given by

$$\boldsymbol{\mu} = \mathbf{K}_{\mathbf{x},\mathbf{t}}(\mathbf{K}_{\mathbf{t},\mathbf{t}} + \sigma^2 \mathbf{I})^{-1}\mathbf{v}, \tag{1}$$

$$\boldsymbol{\Sigma} = \mathbf{K}_{\mathbf{x},\mathbf{x}} - \mathbf{K}_{\mathbf{x},\mathbf{t}}(\mathbf{K}_{\mathbf{t},\mathbf{t}} + \sigma^2 \mathbf{I})^{-1}\mathbf{K}_{\mathbf{t},\mathbf{x}} \tag{2}$$

where $\mathbf{K}_{\mathbf{x},\mathbf{t}}$ denotes the covariance matrix with $[\mathbf{K}_{\mathbf{x},\mathbf{t}}]_{ij} = k(x_i, t_j)$. We note that it takes $\mathcal{O}(n^3 + nd)$ time to exactly compute the posterior mean $\boldsymbol{\mu}$, and $\mathcal{O}(n^3 + n^2 d + nd^2)$ time to exactly compute the full posterior covariance matrix $\boldsymbol{\Sigma}$, where $n = |\mathbf{t}|$ and $d = |\mathbf{x}|$.

## 3   The GP adapter and uncertainty-aware time series classification

In this section we describe our framework for time series classification in the presence of sparse and irregular sampling. Our framework enables any black-box classifier learnable by gradient-based methods to be applied to the problem of classifying sparse and irregularly sampled time series.

### 3.1   Classification frameworks and the Gaussian process adapter

In Section 2 we described how we can represent a time series through the marginal posterior it induces under a Gaussian process regression model at any set of reference time points $\mathbf{x}$. By fixing a common

set of reference time points $\mathbf{x}$ for all time series in a data set, every time series can be transformed into a common representation in the form of a multivariate Gaussian $\mathcal{N}(\mathbf{z}|\boldsymbol{\mu}, \boldsymbol{\Sigma}; \boldsymbol{\theta})$ with $\mathbf{z}$ being the random vector distributed according to the posterior GP marginalized over the time points $\mathbf{x}$.[1] Here we assume that the GP parameters $\boldsymbol{\theta}$ are shared across the entire data set.

If the $\mathbf{z}$ values were observed, we could simply apply a black-box classifier. A classifier can be generally defined by a mapping function $f(\mathbf{z}; \mathbf{w})$ parameterized by $\mathbf{w}$, associated with a loss function $\ell(f(\mathbf{z}; \mathbf{w}), y)$ where $y$ is a label value from the output space $\mathcal{Y}$. However, in our case $\mathbf{z}$ is a Gaussian random variable, which means $\ell(f(\mathbf{z}; \mathbf{w}), y)$ is now itself a random variable given a label $y$. Therefore, we use the expectation $\mathbb{E}_{\mathbf{z} \sim \mathcal{N}(\boldsymbol{\mu}, \boldsymbol{\Sigma}; \boldsymbol{\theta})}\big[\ell(f(\mathbf{z}; \mathbf{w}), y)\big]$ as the overall loss between the label $y$ and a time series $\mathcal{S}$ given its Gaussian representation $\mathcal{N}(\boldsymbol{\mu}, \boldsymbol{\Sigma}; \boldsymbol{\theta})$. The learning problem becomes minimizing the expected loss over the entire data set:

$$\mathbf{w}^*, \boldsymbol{\theta}^* = \underset{\mathbf{w}, \boldsymbol{\theta}}{\operatorname{argmin}} \sum_{i=1}^{N} \mathbb{E}_{\mathbf{z}_i \sim \mathcal{N}(\boldsymbol{\mu}_i, \boldsymbol{\Sigma}_i; \boldsymbol{\theta})}\big[\ell(f(\mathbf{z}_i; \mathbf{w}), y_i)\big]. \tag{3}$$

Once we have the optimal parameters $\mathbf{w}^*$ and $\boldsymbol{\theta}^*$, we can make predictions on unseen data. In general, given an unseen time series $\mathcal{S}$ and its Gaussian representation $\mathcal{N}(\boldsymbol{\mu}, \boldsymbol{\Sigma}; \boldsymbol{\theta}^*)$, we can predict its label using (4), although in many cases this can be simplified into a function of $f(\mathbf{z}; \mathbf{w}^*)$ with the expectation taken on or inside of $f(\mathbf{z}; \mathbf{w}^*)$.

$$y^* = \underset{y \in \mathcal{Y}}{\operatorname{argmin}} \, \mathbb{E}_{\mathbf{z} \sim \mathcal{N}(\boldsymbol{\mu}, \boldsymbol{\Sigma}; \boldsymbol{\theta}^*)}\big[\ell(f(\mathbf{z}; \mathbf{w}^*), y)\big] \tag{4}$$

We name the above approach the *Uncertainty-Aware Classification* (UAC) framework. Importantly, this framework propagates the uncertainty in the GP posterior induced by each time series all the way through to the loss function. Besides, we call the transformation $\mathcal{S} \mapsto (\boldsymbol{\mu}, \boldsymbol{\Sigma})$ the *Gaussian process adapter*, since it provides a uniform representation to connect the raw irregularly sampled time series data to a black-box classifier.

Variations of the UAC framework can be derived by taking the expectation at various position of $f(\mathbf{z}; \mathbf{w})$ where $\mathbf{z} \sim \mathcal{N}(\boldsymbol{\mu}, \boldsymbol{\Sigma}; \boldsymbol{\theta})$. Taking the expectation at an earlier stage simplifies the computation, but the uncertainty information will be integrated out earlier as well.[2] In the extreme case, if the expectation is computed immediately followed by the GP adapter transformation, it is equivalent to using a plug-in estimate $\boldsymbol{\mu}$ for $\mathbf{z}$ in the loss function, $\ell(f(\mathbb{E}_{\mathbf{z} \sim \mathcal{N}(\boldsymbol{\mu}, \boldsymbol{\Sigma}; \boldsymbol{\theta})}[\mathbf{z}]; \mathbf{w}), y) = \ell(f(\boldsymbol{\mu}; \mathbf{w}), y)$. We refer to this as the IMPutation (IMP) framework. The IMP framework discards the uncertainty information completely, which further simplifies the computation. This simplified variation may be useful when the time series are more densely sampled, where the uncertainty is less of a concern.

In practice, we can train the model using the UAC objective (3) and predict instead by IMP. In that case, the predictions would be deterministic and can be computed efficiently without drawing samples from the posterior GP as described later in Section 4.

### 3.2 Learning with the GP adapter

In the previous section, we showed that the UAC framework can be trained using (3). In this paper, we use stochastic gradient descent to scalably optimize (3) by updating the model using a single time series at a time, although it can be easily modified for batch or mini-batch updates. From now on, we will focus on the optimization problem $\min_{\mathbf{w}, \boldsymbol{\theta}} \mathbb{E}_{\mathbf{z} \sim \mathcal{N}(\boldsymbol{\mu}, \boldsymbol{\Sigma}; \boldsymbol{\theta})}\big[\ell(f(\mathbf{z}; \mathbf{w}), y)\big]$ where $\boldsymbol{\mu}, \boldsymbol{\Sigma}$ are the output of the GP adapter given a time series $\mathcal{S} = (\mathbf{t}, \mathbf{v})$ and its label $y$. For many classifiers, the expected loss $\mathbb{E}_{\mathbf{z} \sim \mathcal{N}(\boldsymbol{\mu}, \boldsymbol{\Sigma}; \boldsymbol{\theta})}\big[\ell(f(\mathbf{z}; \mathbf{w}), y)\big]$ cannot be analytically computed. In such cases, we use the Monte Carlo average to approximate the expected loss:

$$\mathbb{E}_{\mathbf{z} \sim \mathcal{N}(\boldsymbol{\mu}, \boldsymbol{\Sigma}; \boldsymbol{\theta})}\big[\ell(f(\mathbf{z}; \mathbf{w}), y)\big] \approx \frac{1}{S} \sum_{s=1}^{S} \ell(f(\mathbf{z}_s; \mathbf{w}), y), \quad \text{where } \mathbf{z}_s \sim \mathcal{N}(\boldsymbol{\mu}, \boldsymbol{\Sigma}; \boldsymbol{\theta}). \tag{5}$$

To learn the parameters of both the classifier $\mathbf{w}$ and the Gaussian process regression model $\boldsymbol{\theta}$ jointly under the expected loss, we need to be able to compute the gradient of the expectation given in (5).

To achieve this, we reparameterize the Gaussian random variable using the identity $\mathbf{z} = \boldsymbol{\mu} + \mathbf{R}\boldsymbol{\xi}$ where $\boldsymbol{\xi} \sim \mathcal{N}(\mathbf{0}, \mathbf{I})$ and $\mathbf{R}$ satisfies $\boldsymbol{\Sigma} = \mathbf{R}\mathbf{R}^\top$ [11]. The gradients under this reparameterization are given below, both of which can be approximated using Monte Carlo sampling as in (5). We will focus on efficiently computing the gradient shown in (7) since we assume that the gradient of the base classifier $f(\mathbf{z}; \mathbf{w})$ can be computed efficiently.

$$\frac{\partial}{\partial \mathbf{w}} \mathbb{E}_{\mathbf{z} \sim \mathcal{N}(\boldsymbol{\mu}, \boldsymbol{\Sigma}; \boldsymbol{\theta})} \big[ \ell(f(\mathbf{z}; \mathbf{w}), y) \big] = \mathbb{E}_{\boldsymbol{\xi} \sim \mathcal{N}(\mathbf{0}, \mathbf{I})} \left[ \frac{\partial}{\partial \mathbf{w}} \ell(f(\mathbf{z}; \mathbf{w}), y) \right] \tag{6}$$

$$\frac{\partial}{\partial \boldsymbol{\theta}} \mathbb{E}_{\mathbf{z} \sim \mathcal{N}(\boldsymbol{\mu}, \boldsymbol{\Sigma}; \boldsymbol{\theta})} \big[ \ell(f(\mathbf{z}; \mathbf{w}), y) \big] = \mathbb{E}_{\boldsymbol{\xi} \sim \mathcal{N}(\mathbf{0}, \mathbf{I})} \left[ \sum_i \frac{\partial \ell(f(\mathbf{z}; \mathbf{w}), y)}{\partial z_i} \frac{\partial z_i}{\partial \boldsymbol{\theta}} \right] \tag{7}$$

There are several choices for $\mathbf{R}$ that satisfy $\boldsymbol{\Sigma} = \mathbf{R}\mathbf{R}^\top$. One common choice of $\mathbf{R}$ is the Cholesky factor, a lower triangular matrix, which can be computed using Cholesky decomposition in $\mathcal{O}(d^3)$ for a $d \times d$ covariance matrix $\boldsymbol{\Sigma}$ [7]. We instead use the symmetric matrix square root $\mathbf{R} = \boldsymbol{\Sigma}^{1/2}$. We will show that this particular choice of $\mathbf{R}$ leads to an efficient and scalable approximation algorithm in Section 4.2.

## 4 Fast sampling from posterior Gaussian processes

The computation required by the GP adapter is dominated by the time needed to draw samples from the marginal GP posterior using $\mathbf{z} = \boldsymbol{\mu} + \boldsymbol{\Sigma}^{1/2}\boldsymbol{\xi}$. In Section 2 we noted that the time complexity of exactly computing the posterior mean $\boldsymbol{\mu}$ and covariance $\boldsymbol{\Sigma}$ is $\mathcal{O}(n^3 + nd)$ and $\mathcal{O}(n^3 + n^2 d + nd^2)$, respectively. Once we have both $\boldsymbol{\mu}$ and $\boldsymbol{\Sigma}$ we still need to compute the square root of $\boldsymbol{\Sigma}$, which requires an additional $\mathcal{O}(d^3)$ time to compute exactly. In this section, we show how to efficiently generate samples of $\mathbf{z}$.

### 4.1 Structured kernel interpolation for approximating GP posterior means

The main idea of the structured kernel interpolation (SKI) framework recently proposed by Wilson and Nickisch [25] is to approximate a stationary kernel matrix $\mathbf{K}_{\mathbf{a},\mathbf{b}}$ by the approximate kernel $\widetilde{\mathbf{K}}_{\mathbf{a},\mathbf{b}}$ defined below where $\mathbf{u} = [u_1, \ldots, u_m]^\top$ is a collection of evenly-spaced inducing points.

$$\mathbf{K}_{\mathbf{a},\mathbf{b}} \approx \widetilde{\mathbf{K}}_{\mathbf{a},\mathbf{b}} = \mathbf{W}_{\mathbf{a}} \mathbf{K}_{\mathbf{u},\mathbf{u}} \mathbf{W}_{\mathbf{b}}^\top. \tag{8}$$

Letting $p = |\mathbf{a}|$ and $q = |\mathbf{b}|$, $\mathbf{W}_{\mathbf{a}} \in \mathbb{R}^{p \times m}$ is a sparse interpolation matrix where each row contains only a small number of non-zero entries. We use local cubic convolution interpolation (cubic interpolation for short) [10] as suggested in Wilson and Nickisch [25]. Each row of the interpolation matrices $\mathbf{W}_{\mathbf{a}}, \mathbf{W}_{\mathbf{b}}$ has at most four non-zero entries. Wilson and Nickisch [25] showed that when the kernel is locally smooth (under the resolution of $\mathbf{u}$), cubic interpolation results in accurate approximation. This can be justified as follows: with cubic interpolation, the SKI kernel is essentially the two-dimensional cubic interpolation of $\mathbf{K}_{\mathbf{a},\mathbf{b}}$ using the exact regularly spaced samples stored in $\mathbf{K}_{\mathbf{u},\mathbf{u}}$, which corresponds to classical bicubic convolution. In fact, we can show that $\widetilde{\mathbf{K}}_{\mathbf{a},\mathbf{b}}$ asymptotically converges to $\mathbf{K}_{\mathbf{a},\mathbf{b}}$ as $m$ increases by following the derivation in Keys [10].

Plugging the SKI kernel into (1), the posterior GP mean evaluated at $\mathbf{x}$ can be approximated by

$$\boldsymbol{\mu} = \mathbf{K}_{\mathbf{x},\mathbf{t}} \left( \mathbf{K}_{\mathbf{t},\mathbf{t}} + \sigma^2 \mathbf{I} \right)^{-1} \mathbf{v} \approx \mathbf{W}_{\mathbf{x}} \mathbf{K}_{\mathbf{u},\mathbf{u}} \mathbf{W}_{\mathbf{t}}^\top \left( \mathbf{W}_{\mathbf{t}} \mathbf{K}_{\mathbf{u},\mathbf{u}}^{-1} \mathbf{W}_{\mathbf{t}}^\top + \sigma^2 \mathbf{I} \right)^{-1} \mathbf{v}. \tag{9}$$

The inducing points $\mathbf{u}$ are chosen to be evenly-spaced because $\mathbf{K}_{\mathbf{u},\mathbf{u}}$ forms a symmetric Toeplitz matrix under a stationary covariance function. A symmetric Toeplitz matrix can be embedded into a circulant matrix to perform matrix vector multiplication using fast Fourier transforms [7].

Further, one can use the conjugate gradient method to solve for $(\mathbf{W}_{\mathbf{t}} \mathbf{K}_{\mathbf{u},\mathbf{u}}^{-1} \mathbf{W}_{\mathbf{t}}^\top + \sigma^2 \mathbf{I})^{-1}\mathbf{v}$ which only involves computing the matrix-vector product $(\mathbf{W}_{\mathbf{t}} \mathbf{K}_{\mathbf{u},\mathbf{u}}^{-1} \mathbf{W}_{\mathbf{t}}^\top + \sigma^2 \mathbf{I})\mathbf{v}$. In practice, the conjugate gradient method converges within only a few iterations. Therefore, approximating the posterior mean $\boldsymbol{\mu}$ using SKI takes only $\mathcal{O}(n + d + m \log m)$ time to compute. In addition, since a symmetric Toeplitz matrix $\mathbf{K}_{\mathbf{u},\mathbf{u}}$ can be uniquely characterized by its first column, and $\mathbf{W}_{\mathbf{t}}$ can be stored as a sparse matrix, approximating $\boldsymbol{\mu}$ requires only $\mathcal{O}(n + d + m)$ space.

**Algorithm 1:** Lanczos method for approximating $\boldsymbol{\Sigma}^{1/2}\boldsymbol{\xi}$

---

**Input**: covariance matrix $\boldsymbol{\Sigma}$, dimension of the Krylov subspace $k$, random vector $\boldsymbol{\xi}$
$\beta_1 = 0$ and $\mathbf{d}_0 = \mathbf{0}$
$\mathbf{d}_1 = \boldsymbol{\xi}/\|\boldsymbol{\xi}\|$
**for** $j = 1$ **to** $k$ **do**
$\quad \mathbf{d} = \boldsymbol{\Sigma}\mathbf{d}_j - \beta_j\mathbf{d}_{j-1}$
$\quad \alpha_j = \mathbf{d}_j^\top\mathbf{d}$
$\quad \mathbf{d} = \mathbf{d} - \alpha_j\mathbf{d}_j$
$\quad \beta_{j+1} = \|\mathbf{d}\|$
$\quad \mathbf{d}_{j+1} = \mathbf{d}/\beta_{j+1}$

$$\mathbf{H} = \text{tridiagonal}(\boldsymbol{\beta}, \boldsymbol{\alpha}, \boldsymbol{\beta}) = \begin{bmatrix} \alpha_1 & \beta_2 & & & \\ \beta_2 & \alpha_2 & \beta_3 & & \\ & \beta_3 & \alpha_3 & \ddots & \\ & & \ddots & \ddots & \beta_k \\ & & & \beta_k & \alpha_k \end{bmatrix}$$

$\mathbf{D} = [\mathbf{d}_1, \ldots, \mathbf{d}_k]$
$\mathbf{H} = \text{tridiagonal}(\boldsymbol{\beta}, \boldsymbol{\alpha}, \boldsymbol{\beta})$
**return** $\|\boldsymbol{\xi}\|\mathbf{D}\mathbf{H}^{1/2}\mathbf{e}_1$      // $\mathbf{e}_1 = [1, 0, \ldots, 0]^\top$

## 4.2 The Lanczos method for covariance square root-vector products

With the SKI techniques, although we can efficiently approximate the posterior mean $\boldsymbol{\mu}$, computing $\boldsymbol{\Sigma}^{1/2}\boldsymbol{\xi}$ is still challenging. If computed exactly, it takes $\mathcal{O}(n^3 + n^2 d + nd^2)$ time to compute $\boldsymbol{\Sigma}$ and $\mathcal{O}(d^3)$ time to take the square root. To overcome the bottleneck, we apply the SKI kernel to the Lanczos method, one of the Krylov subspace approximation methods, to speed up the computation of $\boldsymbol{\Sigma}^{1/2}\boldsymbol{\xi}$ as shown in Algorithm 1. The advantage of the Lanczos method is that neither $\boldsymbol{\Sigma}$ nor $\boldsymbol{\Sigma}^{1/2}$ needs to be computed explicitly. Like the conjugate gradient method, another example of the Krylov subspace method, it only requires the computation of matrix-vector products with $\boldsymbol{\Sigma}$ as the matrix.

The idea of the Lanczos method is to approximate $\boldsymbol{\Sigma}^{1/2}\boldsymbol{\xi}$ in the Krylov subspace $K_k(\boldsymbol{\Sigma}, \boldsymbol{\xi}) = \text{span}\{\boldsymbol{\xi}, \boldsymbol{\Sigma}\boldsymbol{\xi}, \ldots, \boldsymbol{\Sigma}^{k-1}\boldsymbol{\xi}\}$. The iteration in Algorithm 1, usually referred to the Lanczos process, essentially performs the Gram-Schmidt process to transform the basis $\{\boldsymbol{\xi}, \boldsymbol{\Sigma}\boldsymbol{\xi}, \ldots, \boldsymbol{\Sigma}^{k-1}\boldsymbol{\xi}\}$ into an orthonormal basis $\{\mathbf{d}_1, \ldots, \mathbf{d}_k\}$ for the subspace $K_k(\boldsymbol{\Sigma}, \boldsymbol{\xi})$.

The optimal approximation of $\boldsymbol{\Sigma}^{1/2}\boldsymbol{\xi}$ in the Krylov subspace $K_k(\boldsymbol{\Sigma}, \boldsymbol{\xi})$ that minimizes the $\ell^2$-norm of the error is the orthogonal projection of $\boldsymbol{\Sigma}^{1/2}\boldsymbol{\xi}$ onto $K_k(\boldsymbol{\Sigma}, \boldsymbol{\xi})$ as $\mathbf{y}^* = \mathbf{D}\mathbf{D}^\top\boldsymbol{\Sigma}^{1/2}\boldsymbol{\xi}$. Since we choose $\mathbf{d}_1 = \boldsymbol{\xi}/\|\boldsymbol{\xi}\|$, the optimal projection can be written as $\mathbf{y}^* = \|\boldsymbol{\xi}\|\mathbf{D}\mathbf{D}^\top\boldsymbol{\Sigma}^{1/2}\mathbf{D}\mathbf{e}_1$ where $\mathbf{e}_1 = [1, 0, \ldots, 0]^\top$ is the first column of the identify matrix.

One can show that the tridiagonal matrix $\mathbf{H}$ defined in Algorithm 1 satisfies $\mathbf{D}^\top\boldsymbol{\Sigma}\mathbf{D} = \mathbf{H}$ [20]. Also, we have $\mathbf{D}^\top\boldsymbol{\Sigma}^{1/2}\mathbf{D} \approx (\mathbf{D}^\top\boldsymbol{\Sigma}\mathbf{D})^{1/2}$ since the eigenvalues of $\mathbf{H}$ approximate the extremal eigenvalues of $\boldsymbol{\Sigma}$ [19]. Therefore we have $\mathbf{y}^* = \|\boldsymbol{\xi}\|\mathbf{D}\mathbf{D}^\top\boldsymbol{\Sigma}^{1/2}\mathbf{D}\mathbf{e}_1 \approx \|\boldsymbol{\xi}\|\mathbf{D}\mathbf{H}^{1/2}\mathbf{e}_1$.

The error bound of the Lanczos method is analyzed in Ilić et al. [9]. Alternatively one can show that the Lanczos approximation converges superlinearly [16]. In practice, for a $d \times d$ covariance matrix $\boldsymbol{\Sigma}$, the approximation is sufficient for our sampling purpose with $k \ll d$. As $\mathbf{H}$ is now a $k \times k$ matrix, we can use any standard method to compute its square root in $\mathcal{O}(k^3)$ time [2], which is considered $\mathcal{O}(1)$ when $k$ is chosen to be a small constant. Now the computation of the Lanczos method for approximating $\boldsymbol{\Sigma}^{1/2}\boldsymbol{\xi}$ is dominated by the matrix-vector product $\boldsymbol{\Sigma}\mathbf{d}$ during the Lanczos process.

Here we apply the SKI kernel trick again to efficiently approximate $\boldsymbol{\Sigma}\mathbf{d}$ by

$$\boldsymbol{\Sigma}\mathbf{d} \approx \mathbf{W_x}\mathbf{K_{u,u}}\mathbf{W_x}^\top\mathbf{d} - \mathbf{W_x}\mathbf{K_{u,u}}\mathbf{W_t}^\top\left(\mathbf{W_t}\mathbf{K_{u,u}}\mathbf{W_t}^\top + \sigma^2\mathbf{I}\right)^{-1}\mathbf{W_t}\mathbf{K_{u,u}}\mathbf{W_x}^\top\mathbf{d}. \tag{10}$$

Similar to the posterior mean, $\boldsymbol{\Sigma}\mathbf{d}$ can be approximated in $\mathcal{O}(n + d + m\log m)$ time and linear space. Therefore, for $k = \mathcal{O}(1)$ basis vectors, the entire Algorithm 1 takes $\mathcal{O}(n + d + m\log m)$ time and $\mathcal{O}(n + d + m)$ space, which is also the complexity to draw a sample from the posterior GP.

To reduce the variance when estimating the expected loss (5), we can draw multiple samples from the posterior GP: $\{\boldsymbol{\Sigma}^{1/2}\boldsymbol{\xi}_s\}_{s=1,\ldots,S}$ where $\boldsymbol{\xi}_s \sim \mathcal{N}(\mathbf{0}, \mathbf{I})$. Since all of the samples are associated with the same covariance matrix $\boldsymbol{\Sigma}$, we can use the block Lanczos process [8], an extension to the single-vector Lanczos method presented in Algorithm 1, to simultaneously approximate $\boldsymbol{\Sigma}^{1/2}\boldsymbol{\Xi}$ for all $S$ random

vectors $\boldsymbol{\Xi} = [\boldsymbol{\xi}_1, \ldots, \boldsymbol{\xi}_S]$. Similarly, during the block Lanczos process, we use the block conjugate gradient method [6, 5] to simultaneously solve the linear equation $(\mathbf{W_t}\mathbf{K_{u,u}}\mathbf{W_t^\top} + \sigma^2\mathbf{I})^{-1}\boldsymbol{\alpha}$ for multiple $\boldsymbol{\alpha}$.

## 5 End-to-end learning with the GP adapter

The most common way to train GP parameters is through maximizing the marginal likelihood [17]

$$\log p(\mathbf{v}|\mathbf{t}, \boldsymbol{\theta}) = -\frac{1}{2}\mathbf{v}^\top \left(\mathbf{K_{t,t}} + \sigma^2\mathbf{I}\right)^{-1}\mathbf{v} - \frac{1}{2}\log\left|\mathbf{K_{t,t}} + \sigma^2\mathbf{I}\right| - \frac{n}{2}\log 2\pi. \qquad (11)$$

If we follow this criterion, training the UAC framework becomes a two-stage procedure: first we learn GP parameters by maximizing the marginal likelihood. We then compute $\boldsymbol{\mu}$ and $\boldsymbol{\Sigma}$ given each time series $\mathcal{S}$ and the learned GP parameters $\boldsymbol{\theta}^*$. Both $\boldsymbol{\mu}$ and $\boldsymbol{\Sigma}$ are then fixed and used to train the classifier using (6).

In this section, we describe how to instead train the GP parameters discriminatively end-to-end using backpropagation. As mentioned in Section 3, we train the UAC framework by jointly optimizing the GP parameters $\boldsymbol{\theta}$ and the parameters of the classifier $\mathbf{w}$ according to (6) and (7).

The most challenging part in (7) is to compute $\partial\mathbf{z} = \partial\boldsymbol{\mu} + \partial(\boldsymbol{\Sigma}^{1/2}\boldsymbol{\xi})$.[3] For $\partial\boldsymbol{\mu}$, we can derive the gradient of the approximating posterior mean (9) as given in Appendix A. Note that the gradient $\partial\boldsymbol{\mu}$ can be approximated efficiently by repeatedly applying fast Fourier transforms and the conjugate gradient method in the same time and space complexity as computing (9).

On the other hand, $\partial(\boldsymbol{\Sigma}^{1/2}\boldsymbol{\xi})$ can be approximated by backpropagating through the Lanczos method described in Algorithm 1. To carry out backpropagation, all operations in the Lanczos method must be differentiable. For the approximation of $\boldsymbol{\Sigma}\mathbf{d}$ during the Lanczos process, we can similarly compute the gradient of (10) efficiently using the SKI techniques as in computing $\partial\boldsymbol{\mu}$ (see Appendix A).

The gradient $\partial\mathbf{H}^{1/2}$ for the last step of Algorithm 1 can be derived as follows. From $\mathbf{H} = \mathbf{H}^{1/2}\mathbf{H}^{1/2}$, we have $\partial\mathbf{H} = (\partial\mathbf{H}^{1/2})\mathbf{H}^{1/2} + \mathbf{H}^{1/2}(\partial\mathbf{H}^{1/2})$. This is known as the Sylvester equation, which has the form of $\mathbf{AX} + \mathbf{XB} = \mathbf{C}$ where $\mathbf{A}, \mathbf{B}, \mathbf{C}$ are matrices and $\mathbf{X}$ is the unknown matrix to solve for. We can compute the gradient $\partial\mathbf{H}^{1/2}$ by solving the Sylvester equation using the Bartels-Stewart algorithm [1] in $\mathcal{O}(k^3)$ time for a $k \times k$ matrix $\mathbf{H}$, which is considered $\mathcal{O}(1)$ for a small constant $k$.

Overall, training the GP adapter using stochastic optimization with the aforementioned approach takes $\mathcal{O}(n + d + m\log m)$ time and $\mathcal{O}(n + d + m)$ space for $m$ inducing points, $n$ observations in the time series, and $d$ features generated by the GP adapter.

## 6 Related work

The recently proposed mixtures of expected Gaussian kernels (MEG) [13] for classification of irregular time series is probably the closest work to ours. The random feature representation of the MEG kernel is in the form of $\sqrt{2/m}\,\mathbb{E}_{\mathbf{z}\sim\mathcal{N}(\boldsymbol{\mu},\boldsymbol{\Sigma})}\left[\cos(\mathbf{w}_i^\top\mathbf{z} + b_i)\right]$, which the algorithm described in Section 4 can be applied to directly. However, by exploiting the spectral property of Gaussian kernels, the expected random feature of the MEG kernel is shown to be analytically computable by $\sqrt{2/m}\,\exp(-\mathbf{w}_i^\top\boldsymbol{\Sigma}\mathbf{w}_i/2)\cos(\mathbf{w}_i^\top\boldsymbol{\mu} + b_i)$. With the SKI techniques, we can efficiently approximate both $\mathbf{w}_i^\top\boldsymbol{\Sigma}\mathbf{w}_i$ and $\mathbf{w}_i^\top\boldsymbol{\mu}$ in the same time and space complexity as the GP adapter. Moreover, the random features of the MEG kernel can be viewed as a stochastic layer in the classification network, with no trainable parameters. All $\{\mathbf{w}_i, b_i\}_{i=1,\ldots,m}$ are randomly initialized once in the beginning and associated with the output of the GP adapter in a nonlinear way described above.

Moreover, the MEG kernel classification is originally a two-stage method: one first estimates the GP parameters by maximizing the marginal likelihood and then uses the optimized GP parameters to compute the MEG kernel for classification. Since the random feature is differentiable, with the approximation of $\partial\boldsymbol{\mu}$ and $\partial(\boldsymbol{\Sigma}\mathbf{d})$ described in Section 5, we can form a similar classification network that can be efficiently trained end-to-end using the GP adapter. In Section 7.2, we will show that training the MEG kernel end-to-end leads to better classification performance.

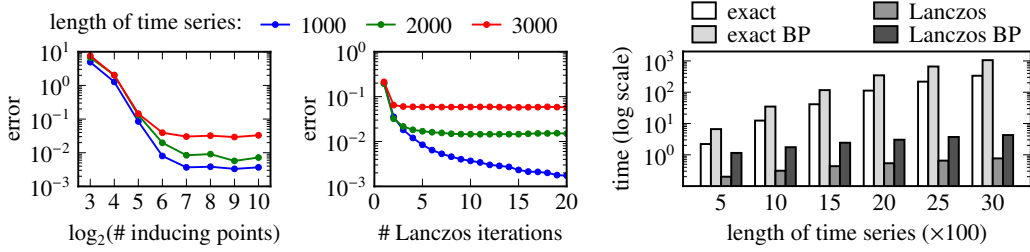

Figure 1: Left: Sample approximation error versus the number of inducing points. Middle: Sample approximation error versus the number of Lanczos iterations. Right: Running time comparisons (in seconds). BP denotes computing the gradient of the sample using backpropagation.

## 7 Experiments

In this section, we present experiments and results exploring several facets of the GP adapter framework including the quality of the approximations and the classification performance of the framework when combined with different base classifiers.

### 7.1 Quality of GP sampling approximations

The key to scalable learning with the GP adapter relies on both fast and accurate approximation for drawing samples from the posterior GP. To assess the approximation quality, we first generate a synthetic sparse and irregularly-sampled time series $\mathcal{S}$ by sampling from a zero-mean Gaussian process at random time points. We use the squared exponential kernel $k(t_i, t_j) = a \exp(-b(t_i - t_j)^2)$ with randomly chosen hyperparameters. We then infer $\boldsymbol{\mu}$ and $\boldsymbol{\Sigma}$ at some reference $\mathbf{x}$ given $\mathcal{S}$. Let $\widetilde{\mathbf{z}}$ denote our approximation of $\mathbf{z} = \boldsymbol{\mu} + \boldsymbol{\Sigma}^{1/2}\boldsymbol{\xi}$. In this experiment, we set the output size $\mathbf{z}$ to be $|\mathcal{S}|$, that is, $d = n$. We evaluate the approximation quality by assessing the error $\|\widetilde{\mathbf{z}} - \mathbf{z}\|$ computed with a fixed random vector $\boldsymbol{\xi}$.

The leftmost plot in Figure 1 shows the approximation error under different numbers of inducing points $m$ with $k = 10$ Lanczos iterations. The middle plot compares the approximation error as the number of Lanczos iterations $k$ varies, with $m = 256$ inducing points. These two plots show that the approximation error drops as more inducing points and Lanczos iterations are used. In both plots, the three lines correspond to different sizes for $\mathbf{z}$: 1000 (bottom line), 2000 (middle line), 3000 (top line). The separation between the curves is due to the fact that the errors are compared under the same number of inducing points. Longer time series leads to lower resolution of the inducing points and hence the higher approximation error.

Note that the approximation error comes from both the cubic interpolation and the Lanczos method. Therefore, to achieve a certain normalized approximation error across different data sizes, we should simultaneously use more inducing points and Lanczos iterations as the data grows. In practice, we find that $k \geq 3$ is sufficient for estimating the expected loss for classification.

The rightmost plot in Figure 1 compares the time to draw a sample using exact computation versus the approximation method described in Section 4 (exact and Lanczos in the figure). We also compare the time to compute the gradient with respect to the GP parameters by both the exact method and the proposed approximation (exact BP and Lanczos BP in the figure) because this is the actual computation carried out during training. In this part of the experiment, we use $k = 10$ and $m = 256$. The plot shows that Lanczos approximation with the SKI kernel yields speed-ups of between 1 and 3 orders of magnitude. Interestingly, for the exact approach, the time for computing the gradient roughly doubles the time of drawing samples. (Note that time is plotted in log scale.) This is because computing gradients requires both forward and backward propagation, whereas drawing samples corresponds to only the forward pass. Both the forward and backward passes take roughly the same computation in the exact case. However, the gap is relatively larger for the approximation approach due to the recursive relationship of the variables in the Lanczos process. In particular, $\mathbf{d}_j$ is defined recursively in terms of all of $\mathbf{d}_1, \ldots, \mathbf{d}_{j-1}$, which makes the backpropagation computation more complicated than the forward pass.

Table 1: Comparison of classification accuracy (in percent). IMP and UAC refer to the loss functions for training described in Section 3.1, and we use IMP predictions throughout. Although not belonging to the UAC framework, we put the MEG kernel in UAC since it is also uncertainty-aware.

|  |  | LogReg | MLP | ConvNet | MEG kernel |
|---|---|---|---|---|---|
| Marginal likelihood | IMP | 77.90 | 85.49 | 87.61 | – |
|  | UAC | 78.23 | 87.05 | 88.17 | 84.82 |
| End-to-end | IMP | 79.12 | 86.49 | 89.84 | – |
|  | UAC | **79.24** | **87.95** | **91.41** | **86.61** |

## 7.2 Classification with GP adapter

In this section, we evaluate the performance of classifying sparse and irregularly-sampled time series using the UAC framework. We test the framework on the uWave data set,[4] a collection of gesture samples categorized into eight gesture patterns [14]. The data set has been split into 3582 training instances and 896 test instances. Each time series contains 945 fully observed samples. Following the data preparation procedure in the MEG kernel work [13], we randomly sample 10% of the observations from each time series to simulate the sparse and irregular sampling scenario. In this experiment, we use the squared exponential covariance function $k(t_i, t_j) = a \exp(-b(t_i - t_j)^2)$ for $a, b > 0$. Together with the independent noise parameter $\sigma^2 > 0$, the GP parameters are $\{a, b, \sigma^2\}$. To bypass the positive constraints on the GP parameters, we reparameterize them by $\{\alpha, \beta, \gamma\}$ such that $a = e^\alpha$, $b = e^\beta$, and $\sigma^2 = e^\gamma$.

To demonstrate that the GP adapter is capable of working with various classifiers, we use the UAC framework to train three different classifiers: a multi-class logistic regression (LogReg), a fully-connected feedforward network (MLP), and a convolutional neural network (ConvNet). The detailed architecture of each model is described in Appendix C.

We use $m = 256$ inducing points, $d = 254$ features output by the GP adapter, $k = 5$ Lanczos iterations, and $S = 10$ samples. We split the training set into two partitions: 70% for training and 30% for validation. We jointly train the classifier with the GP adapter using stochastic gradient descent with Nesterov momentum. We apply early stopping based on the validation set. We also compare to classification with the MEG kernel implemented using our GP adapter as described in Section 6. We use 1000 random features trained with multi-class logistic regression.

Table 1 shows that among all three classifiers, training GP parameters discriminatively always leads to better accuracy than maximizing the marginal likelihood. This claim also holds for the results using the MEG kernel. Further, taking the uncertainty into account by sampling from the posterior GP always outperforms training using only the posterior means. Finally, we can also see that the classification accuracy improves as the model gets deeper.

## 8 Conclusions and future work

We have presented a general framework for classifying sparse and irregularly-sampled time series and have shown how to scale up the required computations using a new approach to generating approximate samples. We have validated the approximation quality, the computational speed-ups, and the benefit of the proposed approach relative to existing baselines.

There are many promising directions for future work including investigating more complicated covariance functions like the spectral mixture kernel [24], different classifiers including the encoder LSTM [23], and extending the framework to multi-dimensional time series and GPs with multi-dimensional index sets (e.g., for spatial data). Lastly, the GP adapter can also be applied to other problems such as dimensionality reduction by combining it with an autoencoder.

**Acknowledgements**

This work was supported by the National Science Foundation under Grant No. 1350522.

## Footnotes

[1] The notation $\mathcal{N}(\boldsymbol{\mu}, \boldsymbol{\Sigma}; \boldsymbol{\theta})$ explicitly expresses that both $\boldsymbol{\mu}$ and $\boldsymbol{\Sigma}$ are functions of the GP parameters $\boldsymbol{\theta}$. Besides, they are also functions of $\mathcal{S} = (\mathbf{t}, \mathbf{v})$ as shown in (1) and (2).

[2] For example, the loss of the expected output of the classifier $\ell(\mathbb{E}_{\mathbf{z} \sim \mathcal{N}(\boldsymbol{\mu}, \boldsymbol{\Sigma}; \boldsymbol{\theta})}[f(\mathbf{z}; \mathbf{w})], y)$.

[3] For brevity, we drop $1/\partial\boldsymbol{\theta}$ from the gradient notation in this section.

[4] The data set `UWaveGestureLibraryAll` is available at `http://timeseriesclassification.com`.

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
