[Supplementary Material · appendix.pdf]

# Appendix for a scalable end-to-end Gaussian process adapter for time series classification

## A  Gradients for GP approximation

### A.1  Gradients of the approximate posterior GP covariance-vector product

Throughout we denote the independent noise variance $\sigma^2$ as $\rho$ for clarity. Let $\widetilde{\boldsymbol{\Sigma}}$ be the approximate posterior covariance derived by the SKI kernel, and $\theta$ be one of the GP hyperparameters. For any vector $\mathbf{d}$, the gradient $\partial\widetilde{\boldsymbol{\Sigma}}\mathbf{d}/\partial\theta$ is given below. Note that during the Lanczos process, $\mathbf{d}$ is a function of $\theta$, which should be properly handled in backpropagation.

$$
\begin{aligned}
&\frac{\partial}{\partial\theta}\left\{\mathbf{W_x}\mathbf{K_{u,u}}\mathbf{W_x^\top}\mathbf{d} - \mathbf{W_x}\mathbf{K_{u,u}}\mathbf{W_t^\top}\left(\mathbf{W_t}\mathbf{K_{u,u}}\mathbf{W_t^\top} + \rho\mathbf{I}\right)^{-1}\mathbf{W_t}\mathbf{K_{u,u}}\mathbf{W_x^\top}\mathbf{d}\right\} \\
&=\quad \mathbf{W_x}\frac{\partial\mathbf{K_{u,u}}}{\partial\theta}\mathbf{W_x^\top}\mathbf{d} \\
&\quad -\mathbf{W_x}\frac{\partial\mathbf{K_{u,u}}}{\partial\theta}\mathbf{W_t^\top}\left(\mathbf{W_t}\mathbf{K_{u,u}}\mathbf{W_t^\top} + \rho\mathbf{I}\right)^{-1}\mathbf{W_t}\mathbf{K_{u,u}}\mathbf{W_x^\top}\mathbf{d} \\
&\quad -\mathbf{W_x}\mathbf{K_{u,u}}\mathbf{W_t^\top}\left(\mathbf{W_t}\mathbf{K_{u,u}}\mathbf{W_t^\top} + \rho\mathbf{I}\right)^{-1}\mathbf{W_t}\frac{\partial\mathbf{K_{u,u}}}{\partial\theta}\mathbf{W_x^\top}\mathbf{d} \\
&\quad +\mathbf{W_x}\mathbf{K_{u,u}}\mathbf{W_t^\top}\left(\mathbf{W_t}\mathbf{K_{u,u}}\mathbf{W_t^\top} + \rho\mathbf{I}\right)^{-1}\mathbf{W_t}\frac{\partial\mathbf{K_{u,u}}}{\partial\theta}\mathbf{W_t^\top} \\
&\quad\quad \left(\mathbf{W_t}\mathbf{K_{u,u}}\mathbf{W_t^\top} + \rho\mathbf{I}\right)^{-1}\mathbf{W_t}\mathbf{K_{u,u}}\mathbf{W_x^\top}\mathbf{d}.
\end{aligned}
$$

To reduce redundant computations, we introduce the following variables:

$$
\begin{aligned}
\boldsymbol{\alpha} &= \mathbf{W_x^\top}\mathbf{d}, \\
\boldsymbol{\beta} &= \frac{\partial\mathbf{K_{u,u}}}{\partial\theta}\boldsymbol{\alpha}, \\
\boldsymbol{\gamma} &= \mathbf{K_{u,u}}\boldsymbol{\alpha}, \\
\boldsymbol{\delta} &= \left(\mathbf{W_t}\mathbf{K_{u,u}}\mathbf{W_t^\top} + \rho\mathbf{I}\right)^{-1}\mathbf{W_t}\boldsymbol{\gamma}, \\
\boldsymbol{\zeta} &= \frac{\partial\mathbf{K_{u,u}}}{\partial\theta}\mathbf{W_t^\top}\boldsymbol{\delta}, \\
\boldsymbol{\eta} &= \mathbf{K_{u,u}}\mathbf{W_t^\top}\left(\mathbf{W_t}\mathbf{K_{u,u}}\mathbf{W_t^\top} + \rho\mathbf{I}\right)^{-1}\mathbf{W_t}\left(\boldsymbol{\zeta} - \boldsymbol{\beta}\right).
\end{aligned}
$$

The gradient with respect to $\theta$ is therefore $\partial\widetilde{\boldsymbol{\Sigma}}\mathbf{d}/\partial\theta = \mathbf{W_x}\left(\boldsymbol{\beta} - \boldsymbol{\zeta} + \boldsymbol{\eta}\right)$.

The gradient $\partial\widetilde{\boldsymbol{\Sigma}}\mathbf{d}/\partial\rho$ with respect to the noise variance $\rho$ is given by

$$
\begin{aligned}
&\frac{\partial}{\partial\rho}\left\{\mathbf{W_x}\mathbf{K_{u,u}}\mathbf{W_x^\top} - \mathbf{W_x}\mathbf{K_{u,u}}\mathbf{W_t^\top}\left(\mathbf{W_t}\mathbf{K_{u,u}}\mathbf{W_t^\top} + \rho\mathbf{I}\right)^{-1}\mathbf{W_t}\mathbf{K_{u,u}}\mathbf{W_x^\top}\mathbf{d}\right\} \\
&=\mathbf{W_x}\mathbf{K_{u,u}}\mathbf{W_t^\top}\left(\mathbf{W_t}\mathbf{K_{u,u}}\mathbf{W_t^\top} + \rho\mathbf{I}\right)^{-1}\left(\mathbf{W_t}\mathbf{K_{u,u}}\mathbf{W_t^\top} + \rho\mathbf{I}\right)^{-1}\mathbf{W_t}\mathbf{K_{u,u}}\mathbf{W_x^\top}\mathbf{d} \\
&=\mathbf{W_x}\mathbf{K_{u,u}}\mathbf{W_t^\top}\left(\mathbf{W_t}\mathbf{K_{u,u}}\mathbf{W_t^\top} + \rho\mathbf{I}\right)^{-1}\boldsymbol{\delta}.
\end{aligned}
$$

### A.2 Gradients of the approximate posterior GP mean

Let $\widetilde{\boldsymbol{\mu}}$ denote the approximate posterior mean derived by the SKI kernel. The gradient $\partial\widetilde{\boldsymbol{\mu}}/\partial\theta$ with respect to the GP hyperparameter $\theta$ is given by

$$
\begin{aligned}
&\frac{\partial}{\partial\theta}\mathbf{W_x}\mathbf{K_{u,u}}\mathbf{W_t^\top}\left(\mathbf{W_t}\mathbf{K_{u,u}}\mathbf{W_t^\top}+\rho\mathbf{I}\right)^{-1}\mathbf{v}\\
=\quad&\mathbf{W_x}\frac{\partial\mathbf{K_{u,u}}}{\partial\theta}\mathbf{W_t^\top}\left(\mathbf{W_t}\mathbf{K_{u,u}}\mathbf{W_t^\top}+\rho\mathbf{I}\right)^{-1}\mathbf{v}\\
&-\mathbf{W_x}\mathbf{K_{u,u}}\mathbf{W_t^\top}\left(\mathbf{W_t}\mathbf{K_{u,u}}\mathbf{W_t^\top}+\rho\mathbf{I}\right)^{-1}\mathbf{W_t}\frac{\partial\mathbf{K_{u,u}}}{\partial\theta}\mathbf{W_t^\top}\left(\mathbf{W_t}\mathbf{K_{u,u}}\mathbf{W_t^\top}+\rho\mathbf{I}\right)^{-1}\mathbf{v}.
\end{aligned}
$$

To reduce redundant computations, we introduce the following variables:

$$
\begin{aligned}
\boldsymbol{\alpha}&=\left(\mathbf{W_t}\mathbf{K_{u,u}}\mathbf{W_t^\top}+\rho\mathbf{I}\right)^{-1}\mathbf{v},\\
\boldsymbol{\beta}&=\frac{\partial\mathbf{K_{u,u}}}{\partial\theta}\mathbf{W_t^\top}\boldsymbol{\alpha},\\
\boldsymbol{\gamma}&=\mathbf{K_{u,u}}\mathbf{W_t^\top}\left(\mathbf{W_t}\mathbf{K_{u,u}}\mathbf{W_t^\top}+\rho\mathbf{I}\right)^{-1}\mathbf{W_t}\boldsymbol{\beta}.
\end{aligned}
$$

The gradient with respect to $\theta$ is therefore $\partial\widetilde{\boldsymbol{\mu}}/\partial\theta=\mathbf{W_x}\left(\boldsymbol{\beta}-\boldsymbol{\gamma}\right)$.

The gradient $\partial\widetilde{\boldsymbol{\mu}}/\partial\rho$ with respect to the noise variance $\rho$ is given by

$$
\begin{aligned}
&\frac{\partial}{\partial\rho}\mathbf{W_x}\mathbf{K_{u,u}}\mathbf{W_t^\top}\left(\mathbf{W_t}\mathbf{K_{u,u}}\mathbf{W_t^\top}+\rho\mathbf{I}\right)^{-1}\mathbf{v}\\
=\;&-\mathbf{W_x}\mathbf{K_{u,u}}\mathbf{W_t^\top}\left(\mathbf{W_t}\mathbf{K_{u,u}}\mathbf{W_t^\top}+\rho\mathbf{I}\right)^{-1}\left(\mathbf{W_t}\mathbf{K_{u,u}}\mathbf{W_t^\top}+\rho\mathbf{I}\right)^{-1}\mathbf{v}\\
=\;&-\mathbf{W_x}\mathbf{K_{u,u}}\mathbf{W_t^\top}\left(\mathbf{W_t}\mathbf{K_{u,u}}\mathbf{W_t^\top}+\rho\mathbf{I}\right)^{-1}\boldsymbol{\alpha}.
\end{aligned}
$$

## B  Cubic interpolation in backpropagation

The choice of cubic convolution interpolation proposed by Keys [1] is preferable over other interpolation methods such as spline interpolation when training the GP parameters. If spline interpolation is used to construct the SKI kernel $\widetilde{\mathbf{K}}_{\mathbf{a,b}}$, the interpolation matrix $\mathbf{W_a}$ depends not only on $\mathbf{a}$ and $\mathbf{u}$ but also on the kernel $\mathbf{K_{u,u}}$, which depends on the GP parameters $\boldsymbol{\theta}$. As a result, the gradient $\partial\mathbf{W_a}/\partial\boldsymbol{\theta}$ needs to be computed and thus introduces a huge overhead in backpropagation. On the other hand, the interpolation matrix based on the cubic convolution interpolation depends only on $\mathbf{a}$ and $\mathbf{u}$, which are fixed once the data are given. Therefore, with cubic convolution interpolation, both $\mathbf{W_a}$ and $\mathbf{W_b}$ are constant matrices throughout the entire training process.

## C  Architectures used in the experiment

The architecture of each classifier compared in Section 7.2 are described as follows. The fully-connected network consists of two fully-connected layers, each of which contains 256 units. The convolutional network contains a total of five layers: the first and the third layer are both one-dimensional convolutional layers with four filters of size 5. The second and the fourth layer are one-dimensional max-pooling layers of size 2. The last layer is a fully-connected layer with 256 units. We apply rectified linear activation to all of the convolutional and fully-connected layers. Each classifier takes $d=254$ input features produced by the GP adapter.

## References

[1] Robert G Keys. Cubic convolution interpolation for digital image processing. *Acoustics, Speech and Signal Processing, IEEE Transactions on*, 29(6):1153–1160, 1981.