[Reviews · NeurIPS 2016]

Reviewer 1

Summary

The paper builds a method of classification of irregularly sampled time series. GP regression is applied to the time series, and an efficient method is proposed to draw samples from the posterior functions, to propagate through the classifier. To do this, a regular set of points is defined on which all the samples will be interpolated (with uncertainty), the classifier is trained on the function values at those points, and the expected loss is computed by Monte Carlo over the GPR posteriors.

Qualitative Assessment

Section 1 The standard of writing in this work is good, it was a pleasure to read the paper, and I understood nearly everything on the first pass. I'm still not entirely clear what the MEG method is, though, but I understand it uses Fourier features and a one-time estimate of the kernel parameters (i.e. it's not 'end-to-end'). Section 2 is standard GP background. Fine. Section 3. line 96. Naming your "framework" is rather overkill, no? You're propagating uncertainty in the inputs through to a classifier model. I'm sure similar things have been done for e.g. missing data: you might view your problem as a missing data problem, with GP imputation, if so inclined. It would be nice if you could outline what MEG does, and how it differs from your approach. My understanding is that the main difference between this paper and MEG is that you propagate samples through, and MEG just propagates the mean, is that right? Section 4 is the most technical part of the paper. In section 4.1, the authors recap the SKI framework of Wilson and Nickish. In section 4.2, they show how do compute approximate square-root-matrix vector multiplication using the Lanczos method. I suspect that this procedure may be straightforward to those more versed in numerical linear algebra, but I think it's a neat cross-over with ML, and it does depend a little on combination with the SKI idea to get fast matrix-vector operations inside the Lanczos loop. In section 5, the authors reiterate that they are doing end-to-end learning, and so will attempt to learn the GP parameters alongside the parameters of the classifier. This means backpropagating through the Lanczos operations. Section 6 does not do a very good job of explaining the difference between this work and MEG. At this point, I'm left with the burning question: "is this worth it"? Section 7. Section 7.1 is well done: it attempt to answer the question about the accuracy of the sampling-approximation. Figure 1 : caption should explain what the colors mean, very confusing otherwise. In section 7.2, the method is demonstrated on a single dataset. Disappointingly, the irregular-sampling of the dataset is artificial i.e. data were sub-sampled irregularly. The experiment does attempt to address my main concern: "is it really worth training everything together", and table 1 purports that this is so. The paper is is rather let down by a lack of scientific investigation as to _why_ this must be so. What were the different kernel parameters that were learned using the two methods? Are the GPs smoother under the end-to-end framework? Does the noise get interpreted differently? Maybe we could see a plot of the different imputed time series?! The paper is also let down by failing to provide a motivating use-case. If it was difficult to find an irregularly sampled dataset, what's the point of the paper? For this reason, I've marked the paper down for impact.

Confidence in this Review

2-Confident (read it all; understood it all reasonably well)


Reviewer 2

Summary

The authors consider classification of an irregularly sampled time-series data. This topic is important. E.g., we would like to classify health state of a person and we have a training sample of ECG signals from different persons. These signals can be sampled with different frequencies under different conditions. In order to solve the problem the authors - consider some reference time points, being the same for all time-series, - model a posterior distribution of each time-series in these reference time-points using GP framework, - in order to reduce computational burden approximate the posterior mean using sparse GP approach, - optimize parameters of a classifier and a GP covariance function simultaneously providing end-to-end training, - approximate optimization criterion using Monte-Carlo. In order to sample from the corresponding GP distributions and reduce a computational burden, the Lanczos approach is used.

Qualitative Assessment

Comments: - line 8. «… data to to any black-box …» -> «… data to any black-box …» - line 47. «… of classification performance including using similar architectures…». Looks like a bad wording - line 73. At the end of the sentence the dot should be removed, at the same time after displayed formulas (1) and (2) a comma and a dot should be used. - line 84. Is it a reasonable assumption to assume that the GP parameters are shared across the entire data set? It seems that it is better to assume that these parameters are shared only inside each class. - lines 89. «...as the loss between the label …» -> «…of the loss between the label …»? - line 101. «Variations of the UAC framework can be derived by taking the expectation at various position of …». To be honest, I do not see any other additional positions except to use mean values instead of a random variable z. Why just not to write that either we perform end-to-end learning simultaneously, or use instead of z its mean value without taking into account variance? - lines 140-145. Here it is written something about sparse interpolation matrices W_a and W_b, cubic interpolation etc. I think that here some more explicit description of these matrices should be provided (their physical sense, how do they look like, etc.) for those who are not going to read [24]. This will improve the readability of the paper. - line 189. It seems that instead of $\Sigma\xi$ it should be $\Sigma^{1/2}\xi$ written. - line 201. «We then we compute …» -> «We then compute ..." - line 230. In the formula for an expected random feature of the MEG kernel there is an error: instead of z its mean value $\mu$ should be used. - line 252. «We evaluation the approximation» -> looks like a bad wording - line 18 from the appendix. «If spine interpolation…» -> «If spline interpolation..." Conclusions: - Classification of an irregularly sampled time-series data is an important topic. - I can not say that the proposed method is absolutely novel. In fact, the method was worked out along existing ideas. However, the authors made a big work in order to bring together all that learning methods, approaches to GP approximation, etc. and as a result they apparently provided a good machinery to make further steps in this classification problem. Therefore, I think that this paper can be published in the NIPS proceedings.

Confidence in this Review

3-Expert (read the paper in detail, know the area, quite certain of my opinion)


Reviewer 3

Summary

This paper presents a framework for the classification of sparse and irregularly sampled time series that uses a “Gaussian process adapter” to connect irregularly sampled time series data to any black-box classifier that can be trained using gradient descent. The GP adapter uses Gaussian process regression to produce an uncertainty-aware and uniform representation for sparse and regularly sampled time series, putting the input data into a fixed-dimensional feature space that is compatible with most commonly used classifiers. One of the major challenges of this approach is its computational limitations, as GP inference is prohibitively expensive when dealing with large time series. The authors address this problem by presenting approximate methods for sampling from the GP posterior and for training the GP adapter end-to-end through backpropagation. They evaluate the sensitivity of the approximation error with respect to the number of inducing points (sampling) and the number of Lanczos iterations (training), and also evaluate the speedups achieved through these approximations. Finally, the demonstrate the effectiveness of the approach on the uWave dataset and show that it outperforms the recently proposed mixtures of expected Gaussian kernels (MEG) approach for classifying irregular time series.

Qualitative Assessment

This paper presents a classification framework that can be used for sparse and irregularly sampled time series. The fact that it can be trained end-to-end, rather than in a two-step process (i.e., maximize marginal likelihood, then train classifier), seems to be a major contribution. The paper is well-written, but the technical details are quite challenging to follow. I think that it could be made a bit more accessible if the authors gave more of an indication about which points are important to understand and which points are finer implementation details. For example, when reading the section describing experiments on the uWave dataset, it wasn’t very clear to me what the implication was that the UAC framework slightly outperformed the IMP framework when trained end-to-end. For readers who don’t have much intuition/background, it would be helpful to include more discussion explaining the tradeoffs between the different approaches shown.

Confidence in this Review

1-Less confident (might not have understood significant parts)


Reviewer 4

Summary

The authors propose an efficient approach for classifying irregularly sampled time series, building on structured kernel interpolation and the Lanczos method. Structured kernel interpolation provides an efficient representation of the kernel, and the Lanczos method is combined with this representation for scalable posterior sampling. The authors assess the approximation error in simulations as function of the user-specified parameters, showing reasonable behaviour, and show improved performance over standard classifiers and a recent MEG kernel approach.

Qualitative Assessment

This is an excellent paper. We often have to perform time series classification of this kind, and many of the methods used are not really intended for these applications, ignoring most of the relevant structure in the data. To approach this problem, the authors make very sensible methodological decisions, intelligently building on the very latest advances in the related literature. Furthermore, the paper is clearly written, and the experiments are enlightening. I really appreciated seeing a synthetic experiment representing a controlled environment, which would give us more insights into the algorithmic behaviour. The range of comparisons in the real experiment also give a good sense of the advantages of the proposed approach. The scope of the paper could be extended to provide an efficient way of sampling from full GP posteriors using structured kernel interpolation. Some minor comments: It is curious that you do not use a GP classifier in Table 1. One of the main advantages of a GP in this context is taking into account the correlations between the different time points, which is only implicitly handled here by the imputation. Also, more than one real experiment would be useful to get a better sense of the general applicability of the approach. But I understand there are space constraints. Perhaps some more experiments could be added to the supplementary material. References [23] and [24] were published in ICML 2013 and ICML 2015, respectively.

Confidence in this Review

3-Expert (read the paper in detail, know the area, quite certain of my opinion)


Reviewer 5

Summary

In this paper the authors introduce a new method to perform classification on irregularly sampled time series which can not be tackled by standard classification due to their various number of input dimensions. For a set of irregularly sampled time series, they place a Gaussian process over each trajectory and compute the mean and variance over a fixed set of input points. For each trajectory this leads to a multivariate Gaussian distribution. In order to use any classifier learnable using gradient descent and to take uncertainty of the GP posterior into account, they compute the expected loss by drawing function values from this multivariate Gaussian. As drawing functions from a GP posterior is cubic in the number of data points, they also present two approximations to speed up this process. Finally they show that their end-to-end approach outperforms the closest related method based on the MEG kernel as well as the quality of their approximations.

Qualitative Assessment

# Technical quality I like that they showed the change of the error of their approximation by changing the number of induced points / number of Lanczos iterations. Also the experiments compared to the closest related method are sound and even show that combined with their end-to-end approach leads to a better performance. However, I would like to have seen a more exhaustive empirical analysis on an original irregularly sampled time series instead of artificially downsampled time series. # Novelty This is the first approach for an end-to-end learning method for irregularly sampled time series. The authors show empirically that their method improves the current state-of-the-art in form of the MEG kernel. Furthermore, it can also be combined with the MEG to an end-to-end approach which leads to better classification performance. # Usefulness Classification of irregularly sampled time series is an unsolved problem in different disciplines such as medicine, biology or climate science. This method allows to tackle this problem with advanced classifications algorithm such as convolutional neural networks which can not be applied out of the box yet. # Clarity The paper is clearly written and the mathematical notation is sound. Minor comments / typos: - What are the units of the y axis in Figure 1 left? Seconds? - Line 201 : "We then we compute" -> "Then we compute" ? - Line 252: "We evaluation" -> "We evaluate" - How are the reference time points x chosen?

Confidence in this Review

2-Confident (read it all; understood it all reasonably well)